# Potential Therapeutic Strategies and Substances for Facial Nerve Regeneration Based on Preclinical Studies

**DOI:** 10.3390/ijms22094926

**Published:** 2021-05-06

**Authors:** Myung Chul Yoo, Jinmann Chon, Junyang Jung, Sung Su Kim, Seonhwan Bae, Sang Hoon Kim, Seung Geun Yeo

**Affiliations:** 1Department of Physical Medicine & Rehabilitation, College of Medicine, Kyung Hee University, Seoul 02447, Korea; famousir@naver.com (M.C.Y.); kkangmann@naver.com (J.C.); 2Department of Anatomy and Neurobiology, College of Medicine, Kyung Hee University, Seoul 02447, Korea; jjung@khu.ac.kr; 3Department of Biochemistry and Molecular Biology, School of Medicine, Kyung Hee University, Seoul 02447, Korea; sgskim@khu.ac.kr; 4Department of Otorhinolaryngology, Head & Neck Surgery, College of Medicine, Kyung Hee University, Seoul 02447, Korea; codingisfun@naver.com (S.B.); hoon0700@naver.com (S.H.K.)

**Keywords:** facial nerve, regeneration, therapeutic strategies, recovery

## Abstract

Despite advances in microsurgical technology and an improved understanding of nerve regeneration, obtaining satisfactory results after facial nerve injury remains a difficult clinical problem. Among existing peripheral nerve regeneration studies, relatively few have focused on the facial nerve, particularly how experimental studies of the facial nerve using animal models play an essential role in understanding functional outcomes and how such studies can lead to improved axon regeneration after nerve injury. The purpose of this article is to review current perspectives on strategies for applying potential therapeutic methods for facial nerve regeneration. To this end, we searched Embase, PubMed, and the Cochrane library using keywords, and after applying exclusion criteria, obtained a total of 31 qualifying experimental studies. We then summarize the fundamental experimental studies on facial nerve regeneration, highlighting recent bioengineering studies employing various strategies for supporting facial nerve regeneration, including nerve conduits with stem cells, neurotrophic factors, and/or other therapeutics. Our summary of the methods and results of these previous reports reveal a common feature among studies, showing that various neurotrophic factors arising from injured nerves contribute to a microenvironment that plays an important role in functional recovery. In most cases, histological examinations showed that this microenvironmental influence increased axonal diameter as well as myelination thickness. Such an analysis of available research on facial nerve injury and regeneration represents the first step toward future therapeutic strategies.

## 1. Introduction

Peripheral facial palsy is caused by a variety of disease pathologies, including congenital conditions, Bell’s palsy, Ramsay Hunt syndrome, Guillain-Barre syndrome, Lyme disease, HIV infection, Kawasaki disease, central nervous system disorders, temporal bone fracture, otitis media, Melkersson-Rosenthal syndrome, neoplasms, and malignant tumors of the parotid gland and temporal bone [1]. The facial nerve follows a long and complicated course from the brainstem to the facial musculature. Most otological procedures, as well as parotid and facial surgeries, are centered around the facial nerve, making it vulnerable to iatrogenic injuries [2]. Temporomandibular joint replacement is the most frequent cause of surgically associated iatrogenic facial nerve injury, with oral and maxillofacial surgical procedures accounting for 40% of injuries, resections of head and neck lesions for 25%, otologic procedures for 17%, cosmetic procedures for 11%, and other procedures for 7% [3]. Depending on the severity of the traumatic insult, functional consequences may vary from slight disfigurement to total loss of function [4]. Recovery is related to the severity of trauma as well as the severity of the symptoms. If mechanical disruption of axons is excessive, the patient is more likely to have a longer recovery time and severe functional outcome with sequelae [4]. In cases of severe injury of the facial nerve, preventing extensive edema formation and inhibiting inflammatory responses are expected to shorten recovery time and improve outcome. Neuronal edema with ischemia causes compression of the facial nerve, which can lead to axonal degeneration and subsequent demyelination of the facial nerve [5].

If the nerve has been transected, the recommended treatment is direct end-to-end anastomosis or use of an autologous nerve graft, depending on the extent (i.e., length) of the injury. The main drawback of nerve repair is its inability to guarantee complete functional recovery. For example, axonal misalignment can cause partially reversible neuronal atrophy, which can interfere with the production of neurotrophic factors for accelerated regeneration [6]. For full restoration of nerve function, the facial nerve must regenerate and expand from the facial motor nerve while maintaining its function, and ultimately must connect to the damaged terminal area [7,8]. However, the molecular mechanisms responsible for axonal regeneration and pathfinding after injury are complex, reflecting crosstalk involving axons and glial cells, neurotrophic factors, extracellular matrix molecules and their receptors. Therefore, it is important to closely consider regenerative effects of agents used to treat facial nerve injury before designing new strategies that can improve specific axon reinnervation and prognosis in facial palsy patients. Importantly, analyzing available research on facial nerve injury and regeneration is the first step toward future therapeutic strategies.

To date, most studies in the literature related to peripheral nerve regeneration have employed a rat model of sciatic nerve injury. Here, we limited our focus to experimental studies involving facial nerve injury. Studies investigating facial nerve-regenerating effects of potential therapeutic strategies and/or substances published from January 2000 to January 2021, identified based on search terms by one of the authors (M.C.Y), were retrieved from four electronic databases: (a) SCOPUS, (b) PubMed, (c) EMBASE and (d) the Cochrane Library. Search terms included “facial nerve,” “facial palsy”, and “nerve regeneration”. Experiments using acupuncture and electrical stimulation therapy, research written in languages other than English, and studies investigating difference methods of surgical reconstruction (end-to-side anastomosis vs. end-to-end anastomosis) or neurorrhaphy were excluded. Pilot studies were also excluded in our review. After application of exclusion criteria, a total of 31 qualifying studies were identified, and their methods and results were summarized (Figure 1).

Of a total of 31 papers, 12 were related to stem cell transplantation, two to stromal vascular fractions, 11 to neurotrophic factors, and six to medications. Three surgical techniques were commonly used. In the first, one side of the facial nerve was exposed and the branch or main stem of the facial nerve was selected. Anastomosis was performed following full-thickness incision and injection of a specific substance intraperitoneally or topically. In the second technique, one side of the facial nerve was exposed, and a specific substance was applied after compression injury to the main trunk of the facial nerve. In the third technique, one side of the facial nerve was exposed, and the branch or main trunk of the facial nerve was selected to create a nerve defect. The nerve, vein conduit or autograft is subsequently filled with a specific substance and connected to the proximal and distal stumps of the nerve.

## 2. Overview of Pathophysiology Following Facial Nerve Injury

After facial nerve injury, resulting axonal injuries cause massive death of neurons through immediate necrosis and prolonged apoptosis, leading to permanent loss of function [9]. The normal Wallerian degeneration process starts 24–36 h after the initial injury, after which axons start to disintegrate and growth factors are released by Schwann cells at the distal end [10]. As the process continues, axons start to collapse and Schwann cells lose their myelin sheath. Macrophages are also recruited, and together with Schwann cells, serve to remove degenerated axons and myelin debris [11]. A number of factors have been demonstrated to participate in this process, including pro-apoptosis factors, neurotrophic factors, and growth associated protein (GAP)-43 [12,13,14]. Furthermore, after a few days, the connections of Schwann cells with axons de-differentiate, and Schwann cells proliferate vigorously. Although it is often possible to repair the injured nerve using microsurgical techniques, the results are generally unsatisfactory. Accordingly, researchers have increasingly turned to more effective regenerative methods premised on the idea that delivery of cellular and/or trophic factors to the site of injury will improve functional recovery of the facial nerve after injury [15,16,17]. In addition, among the numerous treatments for promoting facial nerve repair, the most widely used in current preclinical studies are local interventions, such as nerve anastomosis, nerve transplantation, and nerve conduits containing added neurotrophic factors or neural stem cells. Notably, hundreds of secreted molecules are known to play a role in cell differentiation, cytoskeletal reorganization, axonal guidance, and/or angiogenesis.

## 3. Tissue Engineering with Strategies and Substances in Facial Nerve Regeneration

The primary tools used to date for facial nerve regeneration are biological support materials, stem cells and various growth factors [18], which can be delivered in numerous ways. The tissue-engineered artificial nerve is a bridge that serves as a physical and nutritional support for recovery of the injured facial nerve [19]. Therapeutic molecules in media can be microinjected directly into the nerve ending, [20] or they can be suspended in a fibrin matrix that is injected around the repair sites. In conduit repair, stem cells can be injected in the conduit lumen or embedded in the conduit matrix. Further improvements in delivery systems can provide better cell distribution and improve efficiency. Moreover, three-dimensional printing techniques can be used to manufacture the material in desired shapes, customizing the engineered tissue exactly to match the patient’s specific neural defect.

If the facial nerve injury is severe, such as neurotmesis, the nerve does not spontaneously recover, and outcome is poor even after neurorrhaphy. In cases of extensive injury, autologous nerve graft, with or without nerve conduits, is considered the ‘gold-standard’ clinical technique [21]. In recent decades, considerable effort has been devoted to development of various synthetic and natural nerve-guidance conduits for improving peripheral nerve regeneration following injuries caused by small gaps. Synthetic materials have the advantage of excellent mechanical properties that can accelerate the repair process, induce the migration of Schwann cells, promote normal neural structure formation, and repair long neural defects [18]. There are many types of biological scaffolding materials, including artificial synthetic material and new degradable materials. Polyesters, such as polylactic acid (PLA), polycaprolactone (PCL) and polyglycolic acid (PGA), are common synthetic materials used in neurological tissue engineering [22]. With the exception of autograft technology, most initial research used silicone tubes. The use of silicone tubes requires a secondary surgery to remove the tubes, as implanted silicone can cause a variety of side effects, including inflammation, fibrosis, coagulation, and infection [23]. One approach for avoiding a secondary surgery and preventing these side effects is the use of biodegradable neural tubes made of synthetic biodegradable materials, including collagen, polyphosphoester (PPE), PGA, poly-L-lactic acid (PLLA) and poly-DL-lactide-co-glycolide (PLGA) [24,25]. The nerve regeneration-inducing material that fills the nerve conduit is important for increasing the success rate of the regeneration. In this context, various stem cells, including adipose-derived stem cells (ADSCs), olfactory stem cells (OSCs), neural crest stem-like cells (NCSCs) and immature dental pulp stem cells (iDPSCs), have been investigated for their potential ability to increase nerve regeneration capability.

### 3.1. Stem Cells

Tissue-engineering and regenerative-medicine applications of embryonic stem cells may face ethical issues, spurring research in the use of mesenchymal stem cells (MSCs) derived from other tissues, which retain multipotency potential and the ability to differentiate into neurons and Schwann cells and produce neurotrophic factors. During embryogenesis, the neural crest is transformed into the dorsal neural primordium, from which migratory neural crest cells invade almost all tissues, producing a variety of non-neural cells and classic neural cells [26,27]. The oral mucosa/gingival lamina propria contains a small population of neural crest-origin stem cells (NCSCs), including MSCs [28]. After isolation and in vitro culture, these NCSCs can differentiate into classical neural crest cell lineages, such as neurons and Schwann cells, as well as various non-neural cells, such as melanocytes, endoneural fibroblasts, and a subpopulation of bone marrow and dental MSCs [29,30,31]. A number of studies have investigated stem cell transplantation, both in combination with a nerve conduit and alone, demonstrating significant beneficial effects on rat facial nerve regeneration (Table 1).

#### 3.1.1. Olfactory Stem Cells (OSCs)

Human OSCs, which can be easily obtained from the olfactory mucosa, have been studied in combination with biodegradable hydrogels for their potential use in transplantation in a mouse model of facial nerve injury [32]. Biodegradable hydrogels (e.g., MedGel) are often used to protect transplanted cells from the local environment and prolong their survival, thereby increasing the effectiveness of transplanted cells. Consistent with this, in the study referred to above, a comparison of transplanted OSCs alone with OSCs transplanted with hydrogel (OSCs-MedGel) showed that, although OSCs accelerates recovery of injured facial nerves, the effect lasted for only 1 week, whereas transplanted OSCs-MedGel showed prolonged results (>1 week) [32]. Olfactory mucosa-derived OSCs express representative neural stem cell markers and differentiate into neural stem cells. They can also secrete nerve growth factors and several types of cytokines, growth factors, and neurotrophins that promote nerve regeneration. Taken together, these observations indicate that biodegradable hydrogels maintain the environment surrounding OSCs in the injured area and reduce the negative effects of the local environment on facial nerve paralysis. In addition, a recent preclinical study showed that OSC transplantation following facial nerve repair surgery improved the functional recovery of facial movement and reduced synkinesis [33]. Nasal stem cells, which can be easily obtained under local anesthesia and can cross the blood brain barrier, have been used successfully in various animal models [34]. When compared with other adult stem cells from various tissues, OSCs exert a very strong antiapoptotic effect toward non-activated immune effector cells [35]. Therefore, by limiting injury associated inflammation, OSCs may enhance nerve conduction.

#### 3.1.2. Stem Cells from Human Exfoliated Deciduous Teeth (SHED)

Another study also compared an autograft alone with an autograft incorporating stem cells from human exfoliated deciduous teeth (SHED) in a rat model of facial nerve repair following nerve transection [21]. SHED was shown to induce local neoangiogenesis and differentiation into supporting cells such as Schwann cells, and promote axonal regeneration [21]. In addition, cultured SHED have been shown to produce a number of neurotrophic factors, including neural growth factor (NGF), brain-derived neurotrophic factor (BDNF), and glial cell-derived neurotrophic factor (BDNF). In 2008 and 2011, Sasaki et al. conducted a facial nerve regeneration study in rats treated with nerve conduit and SHED, assessing the degree of neural attachment after configuring a 7-mm gap and tubulization with or without SHED [36,37]. In both cases, the SHED-treated group showed more rapid regeneration than the control group.

#### 3.1.3. Adipose-Derived Stem Cells (ADSCs)

ADSCs, which can be harvested easily and repeatedly with minimally invasive liposuction, also have treatment potential. ADSCs, characterized by their mesenchymal tissue lineage, can differentiate into Schwann-like cells that are relatively simple to dissociate and expand in cell culture [38]. Watanabe et al. showed that, after isolation and culture of ADSCs, silicone tube with both undifferentiated and differentiated ADSCs groups were effectively promoted nerve regeneration in mice with facial paralysis [38]. Similarly, silicone tube with Schwann cells group had therapeutic potential in facial nerve regeneration. In all experimental groups containing these cells, the functional outcome, determined using a facial palsy scoring system, was almost that same as that for the autologous nerve graft group.

Schwann cells are responsible for formation and maintenance of the myelin sheath surrounding axons of peripheral nerves. They also provide neurotrophic factors, such as NGF, GDNF and BDNF, and generate a structural and adhesive extracellular matrix that serves as a scaffold for inducing axonal regeneration [39]. In addition, the secretion of multiple growth factors, including vascular endothelial growth factor (VEGF), basic fibroblast growth factor (bFGF) and hepatocyte growth factor (HGF), expressed in undifferentiated ADSCs cultured under hypoxic conditions, may also have contributed to nerve regeneration [40]. MSCs have also shown increased expression of several neurotrophic and growth factors, as well as tissue-related proteins [41].

Compared with adipose-derived stem cells, dedifferentiated adipocytes have been found to contain higher percentages of undifferentiated stem cells and can be produced on a larger scale [42]. Post-transplant dedifferentiated adipocytes have been shown to promote nerve regeneration in patients with facial nerve defects [43]. Dedifferentiated adipocytes can be separated from a small amount of adipocytes before surgery and expanded in culture to yield a sufficient number of cells, as well as being differentiated into Schwann cells. These features suggest that dedifferentiated adipocytes are more useful than cultured Schwann cells for neuronal regeneration in future clinical applications.

#### 3.1.4. Bone Marrow-Derived Stem Cells (BMSCs) and Dental Pulp Cells (DPCs)

In studies on BMSCs, experimental groups containing undifferentiated BMSC showed more favorable functional results than those containing differentiated BMSCs, an outcome that was related to the secretion of the trophic factors indicated above [44]. DPCs have also been investigated for use in PLGA artificial nerve conduits [37], and another study showed that tubulation with DPCs may promote recovery of nerve defects in rats [45]. Cultured dental pulp-derived cells, which also produce neurotrophic factors such as NGF, BDNF and GDNF, also displayed better therapeutic effects on peripheral nerve regeneration [46,47]. In addition, immature human DPCs have been shown to modulate the effects of proinflammatory cytokines, with increased levels of the anti-inflammatory cytokines IL-6 and IL-10 and reduced levels of proinflammatory factors such as IL-2, IL-4, TNF-α, and IFN-γ [48]. Immature human DPCs were also shown to affect the migration of Schwann cells into the target tissue and their subsequent proliferation, resulting in the production of several types of trophic factors, including neurotrophins, providing a favorable microenvironment for regeneration [49].

#### 3.1.5. Gingiva-Derived Mesenchymal Stem Cells (GMSC), Neural Crest Stem-Like Cell (NCSC)

3D bio-printed scaffold-free nerve tissues have been developed to replace autografts in peripheral nerve regeneration. In particular, scaffold-free nerve constructs generated from GMSC spheroids were found to eliminate the risk of existing scaffold-induced foreign body responses [50]. Most importantly, transplantation of these constructs into a segmental defect of rat facial nerve during defective rat facial nerve regeneration showed a beneficial effect, similar to autografts. These findings suggested that GMSCs can represent promising and easily accessible stem cell sources for tissue engineering in neural tissues.

NCSCs were reported to have the ability to differentiate into classical NC cell lineages, such as neurons and Schwann cells, as well as into a variety of non-neural cell lineages, suggesting that NCSCs may be a promising alternative source of stem cells for cell-based therapy [51]. Implantation of a nerve conduit with NCSCs in a rat model of facial nerve defects was found to promote functional regeneration of the damaged nerve. Compared with parental GMSCs, induced NCSCs showed increased expression of NCSC-related genes and strong differentiation into neurons and Schwann-like cells, demonstrating that parental GMSCs as well as NCSCs may aid in neuronal regeneration.

### 3.2. Stromal Vascular Fraction (SVF)

Infusion of pluripotent MSCs and ADSCs has been widely used to promote nerve regeneration, usually accomplished by inserting an existing artificial neural circuit. Recent studies have also investigated the effects of treatment with ADSCs and the stromal vascular fraction (SVF) of adipose tissue on nerve healing and regeneration after peripheral nerve injuries [52]. Experimentally, SVF is obtained by treatment of subcutaneous adipose tissue, a soft tissue that makes up a large part of the body, with collagenase, which separates mature adipocytes from other cell groups. SVF consists of endothelial cells, pericytes, smooth muscle cells, tissue macrophages and lymphocytes. It was recently shown that SVF also contains MSCs, which it is known can be induced to differentiate into cartilage, bone, muscle, and fat when provided the appropriate environment and biologically active substances. Subcultured ADSCs can differentiate into osteoblasts, chondrocytes and smooth muscle cells, and promote tissue regeneration through the secretion of growth factors and cytokines [53,54]. Shimizu et al. created a biodegradable nerve conduit containing ADSCs and SVF and evaluated its ability to regenerate facial nerves in a rat model with 7-mm nerve gaps [52]. They showed that both ADSCs and SVF similarly facilitated remyelinization, as evidenced by increased myelin sheath thickness, and fiber and axon diameter (Table 2). Thus, similar to the results of previous studies, these authors concluded that SVF, like ADSCs, was effective in promoting neural regeneration. Other studies have also shown that the proper concentration of SVF in silicon tubes is also important, demonstrating that silicone tubes containing 1 × 10^5^ SVF cells significantly increased axon dimeter compared with those containing higher (1 × 10^7^) or lower (1 × 10^3^) numbers [55]. Taken together, these observations indicate that not only differentiated MSCs but also substances from undifferentiated cells, can aid the recovery of facial paralysis, showing that growth factors such as VEGF, bFGF, and HGF secreted from ADSCs found in the SVF act as angiogenic and antiapoptotic growth factors, and that the appropriate concentration of SVF is also important (Table 2).

### 3.3. Neurotrophic Factors

Table 3 demonstrates the summary of experimental design and results contained in articles for regenerative effects on facial nerve injuries according to the neurotrophic factor.

#### 3.3.1. Insulin-Like Growth Factor (IGF)

IGF-1 is known to stimulate the growth and development of neurons and glial cells. Moreover, IGF-1 has been shown to promote neurite growth and neuronal survival [2]. Sugiyama et al. demonstrated that IGF-1 treatment preserves axonal order and myelin, and maintains near-normal Schwann cell proliferation [56]. In one experiment, IGF-1 was administered locally to the damaged nerve using a sustained-release, gelatin-based hydrogel (MedGel) [56]. Notably, the complete recovery rate for the IGF-1–treated group was 66.7% compared to 0% for the saline-treated group. In addition, IGF-1 promotes the myelinating phenotype of Schwann cells in dorsal root ganglion neuron/Schwann cell cocultures through the phosphoinositol 3-kinase (PI3K)/Akt signaling pathway [57]. The underlying mechanism may involve IGF-mediated activation of RAS/MAPK and PI3K/Akt signaling pathways, with consequent inhibition of apoptosis of neurons and Schwann cells and promotion of axon growth [58,59,60].

#### 3.3.2. Fibroblast Growth Factor (FGF)

Basic FGF (bFGF), an important nerve regeneration factor, has been shown to exert a strong effect on angiogenesis. bFGF promotes the proliferation of Schwann cells, but the use of exogenous bFGF for in vivo regeneration of the peripheral nerves is limited by its short in vivo half-life [61]. One successful strategy for peripheral nerve regeneration is to induce migration of a sufficient number of Schwann cells from both nerve stumps. Therefore, whether blood vessels can be induced into the tube to feed and maintain the Schwann cells is an important experimental consideration [62]. Matsumine et al. showed that bFGF continuously released from gelatin hydrogel microspheres during the first 2 weeks of nerve regeneration after peripheral nerve damage significantly increased neural regeneration rate and axonal maturation [61]. Additionally, Komobuchi et al. assessed the effects of exogenous bFGF applied to the lesion in two different ways: as a one-shot application or as a continuous application in a hydrogel [63]. bFGF-containing gelatin hydrogels, which continuously release bFGF, were more effective in promoting nerve regeneration than single-dose bFGF.

#### 3.3.3. Glial Derived Neurotrophic Factor (GDNF)

Barras et al. studied the role of GDNF in facial nerve regeneration, demonstrating that GDNF is a potent neurotrophic factor that improves facial nerve regeneration in the peripheral nervous system and promotes neuron survival in the central nervous system [64]. In preclinical studies, these researchers applied GDNF locally onto the distal sutures of autologous nerve grafts in the mandibular branch of the facial nerve and compared immediate and 7-month delayed repair after nerve transection. The immediate repair group showed a tendency towards worse results with GDNF application; in contrast, delayed repair in the presence of GDNF allowed for much better results. This suggests that the addition of exogenous GDNF may be detrimental to immediate recovery after nerve grafts. The reduced regeneration obtained in the case of immediate repair with the addition of exogenous GDNF may be attributable to a too large a concentration of this neurotrophic factor, which, in excess, can be deleterious for nerve regeneration [65]. However, in the case of prolonged denervation (7-month delayed repair groups), the exogenous supply of neurotrophic substances enhances the ability of Schwann cells to play their role, thereby improving peripheral nerve regeneration in the chronically denervated distal nerve stump.

#### 3.3.4. Transforming Growth Factor-β3 (TGF-β3)

The transforming growth factor beta (TGF-β) family consists of multiple isoforms with a wide range of biological activities. Members of the TGF-β subfamily regulate wound healing, inhibit immune responses, maintain the extracellular matrix, and regulate epithelial and endothelial cell growth and differentiation [66]; they also regulate cell fate, growth, proliferation, apoptosis, differentiation, polarity, migration, invasion and adhesion as well as nerve regeneration, and promote glial scar formation [67]. Elevated expression of TGF-β is also observed after peripheral nerve injury. Specifically, Wang et al. investigated the role of TGF-β3, showing that this isoform greatly increased the diameter and nerve conduction velocity of axons and accelerated facial nerve fiber regeneration and myelination, thus promoting repair of the entire facial nerve [58]. The total number and diameter of nerve fibers was significantly increased in the TGF-β3 group, as compared with in the surgical control group (*p* < 0.01).

#### 3.3.5. Platelet-Rich Plasma (PRP)

Platelet rich plasma (PRP) can be prepared from the patient’s own blood by centrifugation [68]. The advantage of PRP is that it can be obtained easily and used promptly during surgery. It also harbors numerous growth factors after nerve injury [69], including various neurotrophic factors such as neurotrophin-3 (NT-3), angiopoietin-1, GDNF, and BDNF [70]. The use of PRP as a source of neurotrophic factors and its potential for facilitating nerve regeneration have been investigated. On such study investigated PRP in the repair of transected facial nerves in albino guinea pigs [69,70]. These studies provided evidence for the potential use of PRP and/or neural-induced human MSCs, showing that each promotes facial nerve regeneration alone and produce a greater beneficial effect when used in combination. They further demonstrated a marked increase in the expression of neurotrophic factors (NGF, bFGF, angiopoietin-1, BDNF, GDNF, and NT-3) in the facial nerve fragment in the PRP-treated group 10 days after surgery. In particular, NT-3 protein was highly expressed in groups treated PRP, neural-induced human MSCs, and the combination of PRP with neural-induced human MSCs, suggesting that PRP and neural-induced human MSCs are rich sources of neurotrophic factors. A histologic evaluation revealed enhanced myelination and increased axon counts after treatment with PRP and/or neural-induced human MSCs, results similar to previous studies [71,72].

#### 3.3.6. Hepatocyte Growth Factor (HGF)

Generally speaking, systemic administration of a growth factor in solution is not expected to be efficacious in tissue regeneration because of the short half-lives of growth factors [73]. In early peripheral nerve regeneration experiments using neurotrophins, these growth factors were embedded in bioresorbable materials and applied directly to the crushed nerve. In an alternative approach, Kato et al. successfully transfected neurotrophins into peripheral nerves using viral vectors [74,75]. Herpes simplex virus expressing nerve growth factor can be injected directly into the nerve and can be retrogradely transported, mimicking the physiological role of endogenously encoded neurotrophins. Transfected Schwann cells continuously secrete HGF, which, among its many known effects, prevents peripheral nerve degeneration and promotes nerve regeneration.

### 3.4. Collagen-Binding Domain NT-3 (CBD-NT-3)

Collagen, an important part of the extracellular matrix, is present in almost every organ in the body and maintains the structure of the tissue. It can also be used as a transporter of neurotrophic factors to maintain nerve cell growth and promote regeneration of damaged rat nerves [76,77]. In addition, the collagen extracellular matrix serves as a scaffold for migration of fibroblasts, blood vessels, and macrophages at the site of injury. Therefore, endogenous collagen at the injury site can be a binding target for CBD-NT-3, which exerts neurotrophic effects. NT-3 has also been shown to prevent neuron death, rescue damaged nerve cells, and protect nerve cell structure and function. Wang et al. demonstrated that CBD-NT-3 may be useful in promoting nerve regeneration after facial nerve injury, showing that CBD-NT-3 treatment suppressed the pathological process and promoted axon regeneration; as a consequence, most mice in the CBD-NT-3 group were fully recovered at 4 weeks [78].

## 4. Medications

Table 4 demonstrates the regenerative effects of several medications related to the facial nerve. Bumetanide, a diuretic that selectively inhibits NKCC-1 (sodium/potassium chloride transport channel protein) and effectively treats edema [79], has been used to treat spinal cord and sciatic nerve injuries as well as cerebral edema [80,81]. Research has shown that bumetanide can restore damaged peripheral nerve cells by reducing edema and accelerating regeneration. After neural trauma, axotomy or ischemia, NKCC-1 levels increase in neurons, thereby activating GABA-mediated nerve depolarization [82]. Activated GABA inhibits axonal regeneration by stimulating the Rho/Rock signaling pathway and p75NTR synthesis, increasing cell death [83,84]. It also inhibits GABA stimulation, Rho/Rock signaling and p75NTR levels, reducing cell death, accelerating regeneration and increasing endogenous BDNF levels [85]. Moreover, co-treatment with dexamethasone, which possesses antiedematous activity, and bumetanide, which blocks the aquaporin 1 receptor and also decreases edema, exerts a synergistic inhibitory effect on the edema that develops after peripheral nerve regeneration and healing [86]. There have also been studies showing that scar tissue impairs the neural blood supply, resulting in ischemia, which prevents regeneration around nerve tissue [87]. Scar tissue may form an intraneural barrier that hinders the regeneration of nerve axons and induces tissue adhesion, thereby inhibiting the neural blood supply and causing ischemia. Severe injury with compressive edema within nerve bundles results in the adhesion of peripheral nerves to surrounding tissues; the functional recovery of these peripheral nerves is negatively impacted by extraneural scarring. Chitosan is a type of natural high molecular weight biological polysaccharide with high biodegradability and biocompatibility. Use of a chitosan conduit combined with sodium hyaluronate gel was shown to prevent the formation of perineal scars of the facial nerve and to promote the recovery of nerve function [88].

The immunosuppressive agent tacrolimus has also shown positive effects on axon regeneration and nerve healing in facial nerve injury [89]. Tacrolimus, which also acts as a neuroregenerative agent for the peripheral nerves, was investigated for its ability to heal iatrogenic cut injuries of rabbit facial nerves. Measurement of the axon diameter of the nerve by electron microscopy showed that axon diameter was significantly greater in the presence than in the absence of tacrolimus.

Clinical trials have also shown that the L-type voltage-gated calcium channel antagonist, nimodipine [90], promotes peripheral facial nerve function after injury following maxillofacial surgery [91], likely by promoting remyelination by Schwann cells. It has been suggested that nimodipine inhibits microglial activation, thereby inhibiting the release of interleukin-1β and other pro-inflammatory molecules that can mediate degeneration of neurons [92,93]. Another possible mechanism that may explain the protective effect of nimodipine involves increased levels of calcium-binding S-100b, a member of the S-100 family that plays a role in the regulation of intracellular calcium homeostasis [94]. Nimodipine-induced increases in S-100b may have a role in promoting remyelination. Therefore, the protective effect of nimodipine following crush injury of the facial nerve may involve a reduction in inflammation.

Studies have shown that increased oxidative stress and decreased antioxidant enzyme activity are the main causes of nerve damage after axonal injury [10,95]. Thymoquinone, an anti-inflammatory and antioxidant phytochemical, has the ability to remove free radicals, thereby protecting cell membranes from lipid peroxidation caused by trauma; it also has a specific affinity for peroxides, and thus can prevent oxidative damage to multiple tissues [96]. Thymoquinone treatment was shown to further increase nerve regeneration, measured as an increase in axon diameter and thickness of the myelin sheath relative to the postoperative amplitude. Therefore, these authors suggested that we place a greater emphasis on drugs that play a role in free radical scavenging ability as well as anti-inflammatory activity compared with drugs in the steroid group.

## 5. Conclusions and Future Perspectives

Clinical and experimental studies conducted to date have investigated a number of potential factors that increase healing and improve functional outcome after traumatic facial nerve injury, as reviewed here. Studies related to facial nerve regeneration have reached a number of shared conclusions. First is the importance of various neurotrophic factors secreted from damaged nerves, which collectively provide a microenvironment that supports the survival or regeneration of axons. Also important are specific growth factors secreted by transplanted stem cells, including NGF, BDNF, GDNF, NT-3, VEGF, HGF and IGF, which act to promote the recovery and regeneration of damaged peripheral facial nerves. Second, the degree of remyelination after nerve injury determines the speed of recovery. A number of recent studies have investigated active regenerative effects of stem cell-base therapy, showing that these stem cells continue to proliferate even after being transferred to damaged nerve tissues and differentiate into necessary cell types under appropriate microenvironmental conditions [20,98]. Importantly, histological examinations performed following transplantation of different stem cells (e.g., ADSCs, BMSCs) have revealed increases in the thickness of myelination as well as axonal dimeters [38,52,99]. Therefore, local filling of nerve conduit lumens with stem cells can provide a suitable microenvironment for regenerating axons. These stem cells also promote nerve regeneration by secreting various growth factors. However, this research is still at a preclinical stage and has not yet been translated to clinical practice. Although stem cell transplantation or simple application of neurotrophic factors has been shown to improve outcomes, these approaches are not yet superior to surgical treatment methods (e.g., end-to-end anastomosis, free nerve graft, and lateral nerve anastomosis).

As highlighted in this review, despite abundant data from the diverse approaches used for facial nerve regeneration, it is difficult to draw definitive conclusions regarding which combinations are most effective. Hence, pre-clinical and clinical studies comparing different types of stem cells or neurotrophic factors are needed in the future.

## Figures and Tables

**Figure 1 ijms-22-04926-f001:**
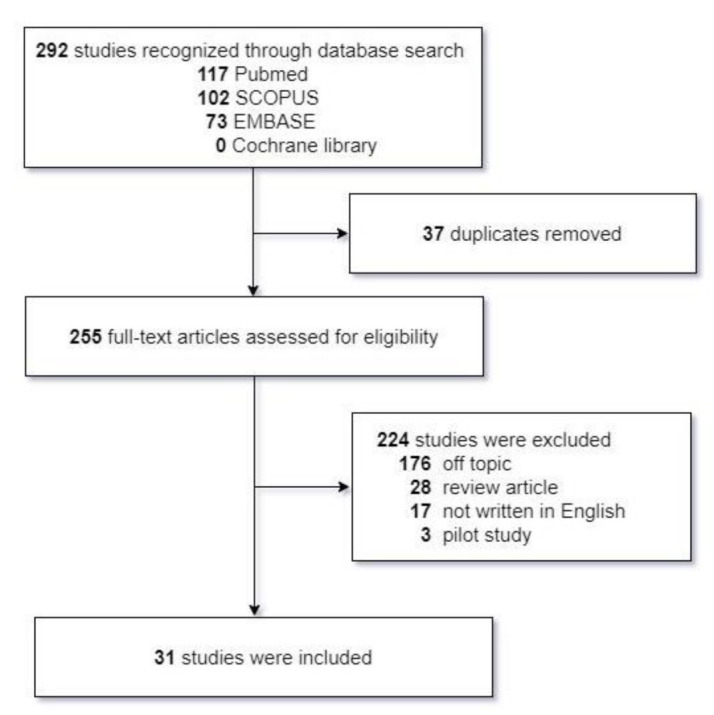
Systematic review of the literature.

**Table 1 ijms-22-04926-t001:** Summary of experimental design and results contained in articles for regenerative effects on facial nerve injuries according to the stem cells.

Potential Substances	Reference	Animal Model	Surgical Procedures	Experimental Design/Therapeutic Molecules	Evaluations	Results	Conclusions
OSC	Bense et al. (2020) [33]	Fisher rats (n = 60)	Rt facial nerve transection (2 mm defect) + femoral vein conduit	Group 1: transection + faciofacial nerve suture only (n = 20) Group 2: femoral vein conduit without OSCs (n = 20) Group 3: femoral vein conduit with OSCs (n = 20)	Facial motor performance: analysis of the interpalpebral distance during the blink reflex Synkinesis: double retrograde labeling of regenerating motoneurons	Maximum amplitude of vibrissae protraction and retraction cycles/angular velocity were increased in Group 3. - OSC transplantation reduced synkinesis.	OSC adjuvant to facial nerve repair surgery improves the functional recovery of facial movement and reduces synkinesis.
DPC	Saez et al. (2019) [48]	Wistar rats (n = 40)	Lt facial nerve compression injury + application of human iDPCs	Group 1: left nerve crushed (n = 20) Group 2: left nerve crushed + iDPCs (n = 20) Group 3: right control for left crushed nerve groups (n = 20)	- Functional recovery: observation of whisker movement - Transmission electron microscopy: nerve morphology - Immunoblotting: NGF expression	- Functional recovery was complete at 14 days in Group 2 but was delayed to 42 days in Group 1. - Group 2 exhibited histological improvement in axons and myelin sheaths. - Group 2 exhibited statistically greater NGF expression.	Human iDPCs promoted regeneration of the facial nerve trunk after 14 days.
SHED	Pereira et al. (2019) [21]	Wistar rats (n = 17)	The buccal branch of the Lt facial nerve transection (5 mm defect) + autograft	Group 1: PGA-collagen nerve conduit with autograft (n = 7) Group 2: PGA-collagen nerve conduit with SHED (autograft, n = 10)	CMAP amplitude: presurgery and 3 and 6 weeks after surgery -Histopathological evaluation: mean axonal density and diameter -Immunofluorescence assays	- Mean CMAP amplitude was higher in Group 2 than in Group 1 (*p* < 0.001). - Mean axonal diameter and axonal density were higher in Group 2 than in Group 1 (*p* = 0.004). - Positive labeling for S100 Schwann-cell marker suggests initiation of differentiation in vivo.	Regeneration was superior in the group treated with SHED
OSC	Esaki et al. (2019) [32]	ICR mice (n = 40)	Rt facial nerve compression injury + OSCs + Megel	Group 1: OSCs + MedGel (n = 10) Group 2: OSCs only (n = 10) Group 3: MedGel (n = 10) Group 4: Mock (DMEM/F-12 alone; n = 10)	-Evaluation of facial nerve paralysis: eye blink, and whisker movement -CMAP amplitude: presurgery and 2 weeks after surgery -Histopathologic evaluation: 1 and 2 weeks after surgery -RT-PCR: neural stem cell markers	- Recovery was more extensive and faster in Group 1. - Nerve function and the number of regenerated nerve fibers were increased in Group 1	OSC-impregnated biodegradable hydrogels produced the most prominent effect on facial nerve recovery.
GMSC	Zhang et al. (2018) [50]	Sprague-Dawley rats (n = 12)	The buccal branch of the Lt facial nerve transection (5 mm defect) + 3D bio-printed nerve constructs.	Group 1: silicon tube control (n = 4) Group 2: autograft (n = 4) Group 3: 3D bio-printed grafts containing human GMSCs (n = 4)	-Facial functional analysis: 12 weeks after surgery-CMAP amplitude -Histological evaluation -Immunohistochemical studies	- Facial palsy score was highest in Group 2 and was higher in Group 3 than in Group 1. - CAMP recovery at 12 weeks and organized axonal alignment were similar in Groups 1 and 2.	3D bio-printed scaffold-free nerve constructs containing GMSC spheroids showed promising beneficial effects on the regeneration of damaged rat facial nerves.
NCSC	Zhang et al. (2018) [51]	Sprague-Dawley rats	The facial nerve transection (6 mm defect) + nerve conduit.	Group 1: controls Group 2: parental GMSCs Group 3: NCSCs	-Facial functional analysis/CMAP amplitude-Electron microscopy: mean axonal density and diameter, myelin thickness -Histological evaluation Immunohistochemical studies	-The induced NCSC population showed increased expression of NCSC-related genes. - NCSCs (Group 3) displayed robust differentiation into neuronal and Schwann-like cells.	Implantation of NCSC-laden nerve conduits promoted functional regeneration of the injured nerve.
DFAT	Matsumine et al. (2014) [43]	Sprague-Dawley rats(n = 25)	The buccal branch of the Lt facial nerve transection (7 mm defect) + silicone tube	Group 1: silicone tube containing type I collagen gel only (n = 7) Group 2: silicone tube containing DFAT (n = 9) Group 3: autologous graft (n = 9)	-CMAP amplitude/latency: 13 weeks after transplantation -Transmission electron microscopy -Immunofluorescence staining	- Axon diameter and myelin thickness were increased and CMAP amplitude was significantly larger in Group 2. -No significant difference between Groups 2 and 3.	DFAT promoted vigorous nerve regeneration.
ADSC	Watanabe et al. (2017) [38]	Lewis rats (n = 77)	The buccal branch of the Lt facial nerve transection (7 mm defect) + silicone tube	Group 1: silicone tube containing uADSCs (n = 16) Group 2: silicone tube containing dADSCs (n = 16) Group 3: silicone tube containing Schwann cells (n = 16) Group 4: silicone tube containing collagen gel alone (n = 16) Group 5: autologous graft (n = 13)	-Facial functional analysis: 13 weeks after transplantation -Transmission electron microscopy -Immunofluorescence staining	- Facial palsy scores were significantly higher in Groups 1, 2, 3, and 5 than in the control group after 6-weeks (*p* < 0.05) and 13-weeks (*p* < 0.001). - Morphometric analyses showed improved regeneration of the nerve in Groups 1–3.	uADSCs and dADSCs may both have therapeutic potential in facial nerve regeneration as a source of Schwann cells in cell-based therapy.
DPC	Sasaki et al. (2014) [45]	Lewis rats (n = 18	The buccal branch of the Lt facial nerve transection (7 mm defect) + silicone tube	Group 1: silicone tube containing collagen gel alone (n = 6) Group 2: autologous nerve graft (n = 6) Group 3: silicone tube containing DPCs (n = 6)	-Facial functional analysis-CMAP amplitude/duration: 13 weeks after transplantation	- Scores in Group 3 were significantly lower than those in the autograft group between 3 and 10 weeks after surgery but were not significantly different at 11 weeks. - CMAP amplitude and duration in Group 3 were not significantly different from those in Group 1 or 2.	Tubulation with DPCs promoted recovery of facial nerve defects and achieved complete recovery comparable to that of nerve autografting in rats.
BMSC	Salomone et al. (2013) [44]	Wistar rats (n = 48)	The mandibular branch of the Rt facial nerve transection (3 mm defect) + silicone tube	Group 1: silicone tube only (n = 12) Group 2: silicone tube containing 200 μL of Matrigel (n = 12) Group 3: silicone tube containing uBMSCs (n = 12) Group 4: silicone tube containing Schwann-like–differentiated cells or dBMSCs (n = 12)	- CMAP amplitude, latency, duration:3 and 6 weeks after surgery - Immunohistochemical staining	- CMAP amplitudes were highest in Groups 3 and 4. - CMAP duration was shorter and distal axonal numbers and density were increased in Group 3.	uBMSC treatment improved facial nerve regeneration.
MSC	Satar et al. (2012) [41]	Sprague-Dawley rats(n = 7)	The buccal branch of both facial nerve transection and anastomosis	Group 1: right anastomosed + MSCs (n = 7) Group 2: left anastomosed-only (n = 7)	RT-PCR Apoptosis assessment	- MSC application increased CNTF, PDGF- α, LIF, TGF- β1, BDNF and NT-3 expression (*p* < 0.05).	MSCs might exert differential effects on tissue-related proteins and trophic/growth factors.
DPC	Sasaki et al. (2011) [37]	Lewis rats (n = 10)	The mandibular branch of both facial nerve transection (7 mm defect) + silicone tube	Group 1: left PLGA tube containing DPCs (n = 10) Group 2: right PLGA tube without DPCs (n = 10)	Immunofluorescence staining Transmission electron microscopy Osmium–toluidine blue-staining	- Nerve repair was more rapid in Group 1 than in Group 2. - Tuj1-positive axons were present in regenerated nerves 2 months after transplantation and no mineralization was detected after 9 weeks.	A PLGA tube filled with DPCs promoted nerve regeneration.

CMAP: Compound muscle action potential; OSC: olfactory stem cell; iDPC: immature dental pulp stem cell; NGF: neural growth factor; SHED: stem cells from human exfoliated deciduous teeth; PGA: polyglycolic acid; ADSC: adipose-derived stem cell; GMSC: gingiva-derived mesenchymal stem cell; NCSC: neural crest stem-like cell; DFAT: dedifferentiated fat cells; uADSC: undifferentiated adipose-derived stem cell; dADSC: differentiated adipose-derived stem cell; uBMSC: undifferentiated BMSClacZ + cell; dBMSC: differentiated BMSClacZ + cell; MSC: mesenchymal stem cell; CNTF: ciliary neurotrophic factor; LIF: leukemia inhibitory factor; TGF-β1: transforming growth factor-β1; BDNF: brain-derived neurotrophic factor; NT-3: neurotrophin-3; PLGA: poly-DL-lactide-co-glycolide; DPC: dental pulp stem cell; qRT-PCR: quantitative reverse transcription polymerase chain reaction.

**Table 2 ijms-22-04926-t002:** Summary of experimental design and results contained in articles for regenerative effects on facial nerve injuries according to the stromal vascular fraction.

Potential Substances	Reference	Animal Model	Surgical Procedures	Experimental Design/Therapeutic Molecules	Evaluations	Results	Conclusions
SVF	Shimizu et al. (2018) [52]	Lewis rats (n = 24)	The buccal branch of the facial nerve transection (7 mm defect) + PGA-collagen nerve conduit	Group 1: PGA-collagen nerve conduit control (n = 8) Group 2: PGA-collagen nerve conduit containing SVF (n = 8) Group 3: PGA-collagen nerve conduit containing ADSCs (n = 8)	CMAP amplitude/latency: 13 weeks after surgery -Histopathologic evaluation Electron microscopy: mean axonal density and diameter, myelin thickness	CMAP amplitude was higher and axon diameter and fiber diameter were larger in Group 2. -Myelin thickness was highest in Group 3.	ADSCs and SVF promote nerve regeneration.
SVF	Matsumine et al. (2017) [55]	Lewis rats (n = 30)	The buccal branch of the Lt facial nerve transection (7 mm defect) + silicone tube + SVF cells	Group 1: autologous graft (n = 6) Group 2: silicone tube containing 1 × 10^3^ SVF cells (n = 6) Group 3: silicone tube containing 1 × 10^5^ SVF cells (n = 6) Group 4: silicone tube containing 1 × 10^7^ SVF cells (n = 6) Group 5: silicone tube containing no SVF cells (n = 6)	-Facial functional analysis -Transmission electron microscopy: 13 weeks after transplantation	- Facial palsy scores were significantly higher in Groups 1 and 3 than in the other groups at 13 weeks after surgery. - Axon diameter and myelin thickness were highest in Group 1 followed by Group 3 (*p* < 0.01).	Infusion of uncultured-SVF into an artificial nerve conduit promoted optimal nerve regeneration.

SVF: stromal vascular fraction; PGA: polyglycolic acid; ADSC: adipose-derived stem cell; CMAP: Compound muscle action potential.

**Table 3 ijms-22-04926-t003:** Summary of experimental design and results contained in articles for regenerative effects on facial nerve injuries according to the neurotrophic factor.

Potential Substances	Reference	Animal Model	Surgical Procedures	Experimental Design/Therapeutic Molecules	Evaluations	Results	Conclusions
IGF-1	Sugiyama et al. (2020) [56]	Hartley guinea pigs (n = 12)	Lt facial nerve compression injury + application of IGF-1	Group 1: saline controls (n = 6) Group 2: IGF-1 (n = 6)	-Eyelid closure/CMAP amplitude: 8 weeks after surgery -Histopathologic evaluation: mean number of myelinated axons -qRT‑PCR: IGF‑1 receptor mRNA levels	-Degree of eyelid closure was greater in Group 2. -Complete recovery rate was greater in Group 2 than Group 1. -CMAP amplitude was positively correlated with the degree of eyelid closure at 8 weeks. IGF-1 receptor mRNA was significantly greater at 7 days after compression than at 2 days.	Topical intratemporal application of IGF-1 produced a significantly higher complete recovery rate.
IGF-1	Bayrak et al. (2017) [2]	New Zealand rabbits (n = 21)	Rt facial nerve crush injury	Group 1:nerve crush injury alone (n = 7) Group 2: nerve crush injury + saline (n = 7) Group 3: nerve crush injury + IGF-1 (n = 7)	-CMAP amplitude: 10 and 42 days after surgery -Histological studies	-CMAP amplitude was significantly lower in Group 2 on day 10 compared with that in Group 3 (*p* < 0.05). -Axonal order and myelin were preserved, and Schwann cell proliferation was close to normal in Group 3 (*p* < 0.05).	Local application of IGF-1 was found to be efficacious in the recovery of a facial nerve crush injury
IGF-1	Matsumine et al. (2016) [61]	Lewis rats (n = 30)	The buccal branch of the Lt facial nerve transection (7 mm defect) + silicone tube	Group 1: silicon tube only (n = 20) Group 2: silicon tube filled with bFGF (n = 10)	-Transmission electron microscopy: mean axonal density and diameter, myelin thickness	-The rate of nerve regeneration and number of regenerating nerve axons was higher in Group 2, which also showed a better degree of maturation of nerve axons.	bFGF was efficacious in promoting facial nerve regeneration.
TGF‑β3	Wang et al. (2016) [58]	Adult rabbits (n = 20)	The buccal branch of the Lt facial nerve transection (5 mm defect) + silicone tube	Group 1: right silicon tube filled with TGF‑β3 (50 ng/μL) (n = 10) Group 2: left silicon tube filled with saline (n = 10) Group 3: surgical control (n = 10).	-CMAP amplitude/CMAP latency: 12 weeks after surgery -Electron microscopy: total number and diameter of regenerated nerve fibers	-The total number and diameter of nerve fibers were significantly increased in the TGF-β3 group, compared with the surgical control group (*p* < 0.01). -Epineurial repair of facial nerves and nerve fibers was complete. -CMAP amplitude was larger and latency was shorter in Group 1.	TGF‑β3 may promote the regeneration of facial nerves.
Neurotrophin-3	Wang et al. (2016) [78]	Sprague-Dawley rats(n = 15)	Lt facial nerve crush injury	Group 1: crush injury + NAT-NT- 3 (n = 5) Group 2: crush injury + CBD-NT-3 (n = 5) Group 3: crush injury + sham (n = 5)	CMAP amplitude Facial nerve function examination Western blotting: NT-3 retention Immunohistochemical staining: collagen content evaluation	-Exogenous NT-3 levels in the CBD-NT-3 group were significantly higher than those in the NAT-NT-3 group. -Axon growth was more ordered and nerve functional recovery was significantly greater in the CBD-NT-3 group than in the NAT-NT-3 group.	CBD-NT-3 enhances facial nerve regeneration and functional recovery.
Hepatocyte growth factor	Esaki et al. (2011) [73]	Balb/C mice (n = 25)	Rt facial nerve crush injury. HSV-HGF, control vector (HSV-LacZ), or medium (PBS) was then applied to the compressed nerve.	Group 1: crush injury + HSV-HGF (n = 5) Group 2: crush injury + HSV-LacZ (n = 5) Group 3: crush injury + PBS (n = 15)	Facial functional analysis CMAP amplitude Enzyme-linked immunosorbent assay: HGF concentration Immunohistochemical staining	-Recovery in the HGF group was significantly faster than that in either the LacZ or PBS group (*p* < 0.01). -Recovery of CMAP amplitude was greater in the HGF group compared with the LacZ group (*p* < 0.01). -The number of myelinated nerve fibers was greater in the HGF group than in the LacZ group.	Introduction of HSV-HGF around the damaged nerve significantly accelerated the recovery of facial nerve function.
bFGF	Komobuchi et al. (2010) [63]	Hartley guinea pigs (n = 24)	Lt facial nerve compression injury + application of bFGF	Group 1: controls (n = 8) Group 2: bFGF single shot (n = 8) Group 3: bFGF-hydrogel (n = 8)	-Evaluation of facial movements: 6 weeks after surgery -Conduction velocity -Histological evaluation	-Facial nerve functional recovery was faster and conduction velocity was greater in Group 3 than in Groups 1 or 2 (*p* < 0.05). -The number of myelinated nerve fibers was significantly larger in Group 3 than in other groups (*p* < 0.05).	A bFGF-impregnated biodegradable hydrogel proved to be effective in facilitating recovery.
PRP and/or MSCs	Cho et al. (2010) [69]	Albino guinea pigs (n = 24)	The Rt facial nerve transection and anastomosis	Group 1: anastomosed only (n = 6) Group 2: anastomosed + PRP (n = 6) Group 3: anastomosed + nMSCs (n = 6) Group 4: anastomosed + PRP + nMSCs (n = 6)	-Facial functional analysis -Electrophysiologic evaluation -Neurotrophic factors assay -Histologic evaluation	-Function and CMAP amplitude were improved in Groups 2–4 compared with the control group 4 weeks after surgery (*p* < 0.05). -Axon counts and myelin thickness were improved in Groups 2–4. -Group 4 had the greatest number of myelinated axon fibers (*p* < 0.05).	PRP and/or nMSCs promote facial nerve regeneration. The combined use of PRP and nMSCs showed a beneficial effect.
GDNF	Barras et al. (2009) [65]	Wistar rats (n = 28)	Immediate and delayed grafts (repair 7 months after the lesion). The buccal branch of the Lt facial nerve transection (10 mm defect) + autologous nerve graft	Group 1: immediate repair, 15-mm autologous graft only (n = 4) Group 2: immediate repair. 12-mm autologous graft + 5-mm channel without GDNF (n = 4) Group 3: immediate repair, 12-mm autologous graft + 5-mm GDNF-releasing channel (n = 6) Group 4: delayed repair, 15-mm autologous graft only (n = 4) Group 5: delayed repair, 12-mm autologous graft + 5-mm channel without GDNF (n = 5) Group 6: delayed repair, 12-mm autologous graft + 5-mm GDNF-releasing channel (n = 5)	-Facial functional analysis: 3 and 6 weeks after nerve repair -Nerve conduction study -Histological analysis: number of myelinated fibers	-GDNF promoted an increase in the number and maturation of nerve fibers, as well as the number of retrogradely labeled neurons in delayed anastomoses.	Application of GDNF to facial nerve grafts via nerve guidance channels improves regeneration after late repair.
PRP	Cho et al. (2009) [70]	Albino guinea pigs (n = 14)	The Rt facial nerve transection and anastomosis	Group 1: controls (n = 7) Group 2: fibrin glue +PRP (n = 7)	-Facial functional analysis -CMAP amplitude-Western blot analysis -Histological evaluation	-High levels of NT-3, angiopoietin-1, GDNF, NGF, and BDNF were observed in Group 2. -Motor function recovery, CMAP amplitude, and axon count were significantly improved in Group 2.	PRP improved functional outcome.
PRP	Farrag et al. (2007) [72]	Sprague-Dawley rats (n = 49)	The buccal branch of the Lt facial nerve transection	Group 1: suture only (n = 11) Group 2: PRP only + no suture (n = 5) Group 3: PRP + suture (n = 5) Group 4: PPP + no suture (n = 5) Group 5: PPP + suture (n = 5) Group 6: fibrin sealant + no suture (n = 12) Group 7: fibrin sealant + suture (n = 6)	-Facial functional analysis -CMAP amplitude/latency/area: presurgery and 8 weeks after surgery -Histomorphometric analysis	-Overall outcomes were improved in the suturing group (*p* < 0.05). -The degree of recovery was greater in Group 2 than Group 4 (*p* < 0.05). -Duration and latency of CMAP and axon counts were most improved in Group 3 compared with suture and PPP-plus-suture groups (*p* < 0.05).	The most favorable results were obtained with PRP added to the suture.

CMAP: Compound muscle action potential; qRT-PCR: quantitative reverse transcription-polymerase chain reaction; IGF-1: insulin-like growth factor 1; NGF: neural growth factor; bFGF: basic fibroblast growth factor; NAT-NT-3: native neurotrophin-3; CBD: collagen-binding domain; MSC: mesenchymal stem cell; BDNF: brain-derived neurotrophic factor; NT-3: neurotrophin-3; HSV-HGF: herpes simplex virus vector that incorporated hepatocyte growth factor; PBS: phosphate-buffered saline; PRP: platelet-rich plasma; PPP: platelet-poor plasma; nMSC: neural-induced mesenchymal stem cell; GDNF: glial cell line-derived neurotrophic factor.

**Table 4 ijms-22-04926-t004:** Summary of experimental design and results contained in articles for regenerative effects on facial nerve injuries according to the medications.

Potential Substances	Reference	Animal Model	Surgical Procedures	Experimental Design/Therapeutic Molecules	Evaluations	Results	Conclusions
Dexamethasone and bumetanide	Longur et al. (2021) [79]	Wistar rats (n = 32)	Rt facial nerve transection and anastomosis	Seven-day treatment Group 1: controls (n = 8) Group 2: bumetanide (15 mg/kg; n = 8) Group 3: dexamethasone (1 mg/kg, intraperitoneally; n = 8) Group 4: bumetanide + dexamethasone (n = 8)	CMAP amplitude/latency: presurgery and 1, 2, and 4 weeks after surgery Histopathologic evaluation: mean number of myelinated axons Western blot analysis: AQP1 band density	-Latency difference in Group 1 was significantly higher than that in Groups 2–4 (*p* = 0.001). -Latency increase in Groups 2 and 3 was higher than that in Group 4 (*p* = 0.002, *p* = 0.046). -The number of myelinated axons was higher in all treatment groups. -Axon number and intensity were higher in group 4 than groups 2 and 3 (*p* = 0.009, *p* = 0.005).	Dexamethasone and bumetanide act synergistically to enhance facial nerve regeneration.
Chitosan	Liu et al. (2018) [88]	New Zealand rabbits (n = 40)	The buccal branch of the Rt facial nerve transection + chitosan conduits or surface-coated with hyaluronate.	Group 1: chitosan only (n = 10) Group 2: chitosan + hyaluronate (n = 10) Group 3: hyaluronate (n = 10) Group 4: controls (n = 10)	-Vibrissae motion evaluation -Scar adhesion analysis-Neural conduction velocity: electrophysiology -Histopathological evaluation: mean number, diameter, and thickness of myelinated axons at 4 and 12 weeks after surgery	-Recovery was greater in Group 2 compared with all other groups. -Group 2 exhibited a greater number of nerve fibers, thicker myelin sheath, and greater nerve conduction velocity.	The use of a chitosan conduit combined with sodium hyaluronate gel may prevent perineural scar formation in facial nerves and promote functional nerve recovery.
Tacrolimus	Tulaci et al. (2016) [89]	New Zealand rabbits (n = 20)	Lt facial nerve transection + anastomosis. Tacrolimus (1 mg/kg/d) was administered subcutaneously for 2 months.	Group 1: controls (n = 10) Group 2: tacrolimus (n = 10)	Electron and light microscopic examinations	-Group 2 showed increased myelinization and thickened endoneurium (axon diameters, thicker myelin sheaths, and higher total number of myelinated axons)	Tacrolimus exerts favorable effects on the healing process of the facial nerve after end-to-end anastomosis.
Thymoquinone	Sereflican et al.(2016) [96]	New Zealand rabbits (n = 24)	The buccal branch of the facial nerve compression injury	Group 1: healthy controls (n = 6) Group 2: crush injury only (n = 6) Group 3: crush injury + thymoquinone (n = 6) Group 4: crush injury + methylprednisolone (n = 6)	CMAP amplitude/latency: pre- and post-surgery at week 8 Histopathological evaluation	-Nerve regeneration was further increased in Group 4 compared with Group 3, as evidenced by increased postoperative CMAP amplitude, axon diameter, and myelin sheath thickness.	Thymoquinone treatment was slightly more efficacious than methylprednisolone treatment in promoting functional nerve recovery.
Nimodipine	Zheng et al. (2015) [90]	Sprague-Dawley rats (n = 63)	The buccal branch of Lt facial nerve crush injury	Group 1: healthy controls (n = 3) Group 2: crush injury only (n = 30) Group 3: crush injury + nimodipine (n = 30)	CMAP amplitude/latency: 3, 10, and 20 days after surgery Facial nerve function examination Immunofluorescence staining	-CMAP amplitude was higher and latency was shorter in Group 3 than in Group 2. - Rats in Group 3 showed clear recovery of myelination and less inflammation compared with those in Group 2. -Staining for S100 calcium-binding protein B was evident in Group 3	Nimodipine treatment ameliorated crush injury damage of the facial nerve in a rat model by promoting remyelination
Etanercept	Topdag et al. (2014) [97]	Wistar albino rats (n = 54)	The facial nerve crush injury Etanercept (6.0 mg/kg) and steroid (1.0 mg/kg) were administered as single intraperitoneal doses immediately after nerve crush.	Group 1: crush injury alone + saline (n = 12) Group 2: crush injury + methylprednisolone (n = 12) Group 3: crush injury + etanercept (n = 12) Group 4: crush injury alone + saline (n = 6) Group 5: crush injury + methylprednisolone (n = 6) Group 6: crush injury + etanercept (n = 6)	Facial functional analysis: 4 and 28 days after surgery Immunohistochemical analysis: macrophage marker, GAP-43 and T cell marker Immunohistochemical staining	-Group 3 showed significantly earlier complete recovery compared with Group 2. -Etanercept and methylprednisolone groups demonstrated a statistically significant difference compared with the control group (*p* < 0.001).	Etanercept treatment accelerated functional recovery after facial nerve crush injury in rats.

CMAP: Compound muscle action potential; AQP1: aquaporin 1; GAP-43: growth-associated protein 43.

## Data Availability

Not applicable.

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
