# Peer review of "Potential Therapeutic Strategies and Substances for Facial Nerve Regeneration Based on Preclinical Studies"

_ijms, 2021, doi:10.3390/ijms22094926_

Round 1
Reviewer 1 Report
In this review, authors discuss current perspectives on strategies for applying potential therapeutic methods for facial nerve regeneration.
A major revision on the following points is requested:
Chapter 1:
- The authors have to better specify what they mean with the sentence “However, the main drawback of nerve repair is that it reduces the production of neurotrophic factors that normally accelerate regeneration; thus, it does not guarantee a functionally complete recovery”. This sentence is not true for all factors involved in the peripheral nerve regeneration process. Which are factors whose expression is reduced after the repair? Which type of repair? How the repair influences neurotrophic factor expression?
- Please, better clarify for which reason 176 papers are considered off topic and which are the common characteristics of the 31 papers selected. Which is the main surgical technique used in the 31 articles selected? How many articles present stem cell transplantation? How many used neurotrophic factors? What is the most widely used animal model for these studies? Authors can decide to address this issue here or in the conclusion.
Chapter 2:
The following sentence need at least one reference: “In addition, among the numerous treatments for promoting facial nerve repair, the most widely used in current clinical practice are local interventions, such as nerve anastomosis, nerve transplantation, and nerve conduits containing added neurotrophic factors or neural stem cells.” My perplexity concerns the facts that conduits enriched with neurotrophic factors and neural stem cells are used in clinic.
Chapter 3:
- Check if the numbering of paragraphs and sub-paragraphs is ok. I suppose that something is wrong.
- Each stem cell type and each factor is discussed singularly. A transversal discussion with author’s opinion about the different stem cells treatment and factors effect is missing. Author’s experience could be reported. Authors can add this discussion point in this chapter or in the conclusion.
- Check lines from 208 to 214. Something went wrong with the formatting of the text.
Chapter 5:
- Lines 391-393: The authors have to specify that these are not resident stem cells, but transplanted stem cells.
- References are missing in the entire paragraph. For example, the sentence “A number of recent studies have investigated active regenerative effects of stem cell-base therapy, showing that these stem cells continue to proliferate even after being transferred to damaged nerve tissues and differentiate into necessary cell types under appropriate microenvironmental conditions.” need a reference. As well as for the sentence “histological examinations performed following transplantation of different stem cells (e.g., ADSCs, BMSCs) have revealed increases in the thickness of myelination as well as axonal dimeters.”
Table:
The authors have to reorganize the table because, as it presented now, it is impossible to consult it. I suggest dividing the table in different tables, for example grouping articles that use the same technique of repair, cell treatment or therapeutic molecules. In the review, tables have to be cited.
References:
References from 84 to 95 are not cited in the text but only in the table. I think that, if the authors selected these articles as important to be reported, they have to comment them in the manuscript.
Author Response
Rebuttal
We would like to thank the Editor and reviewers for their thoughtful, detailed reviews and for providing valuable suggestions for improving our manuscript. We have revised our manuscript in response to these comments; our point-by-point responses to the suggestions of the reviewers are provided below.
* Response to Reviewer 1 comments
Chapter 1:
- Q) The authors have to better specify what they mean with the sentence “However, the main drawback of nerve repair is that it reduces the production of neurotrophic factors that normally accelerate regeneration; thus, it does not guarantee a functionally complete recovery”. This sentence is not true for all factors involved in the peripheral nerve regeneration process. Which are factors whose expression is reduced after the repair? Which type of repair? How the repair influences neurotrophic factor expression?
- A) We agree that, as written, this sentence can be misinterpreted as indicating that, following nerve repair, the production of all neurotrophic factors in the nerve regeneration process is reduced. In general, despite the performance of nerve repair in facial paralysis patients, complete functional recovery cannot be guaranteed. In addition, misalignment of axons can result in partially reversible neuronal atrophy, interfering with the production of neurotrophic factors involved in accelerated regeneration. Therefore, to avoid reader confusion, this sentence has been rewritten.
- Introduction
…
However, the main drawback of nerve repair is that it cannot guarantee complete functional recovery, and axonal misalignment can cause partially reversible neuronal atrophy, which can interfere with the production of neurotrophic factors for accelerated regeneration.
Hussain G, Wang J, Rasul A, Anwar H, Qasim M, Zafar S, Aziz N, Razzaq A, Hussain R, de Aguilar JLG, Sun T. Current Status of Therapeutic Approaches against Peripheral Nerve Injuries: A Detailed Story from Injury to Recovery. Int J Biol Sci 2020; 16(1):116-134.
- Q) Please, better clarify for which reason 176 papers are considered off topic and which are the common characteristics of the 31 papers selected.
- A) To date, most articles related to nerve regeneration after facial paralysis have involved surgical reconstruction. Surgical treatment can be applied clinically to patients with incomplete facial paralysis or complete facial palsy. Many techniques have involved connecting the damaged facial nerve to other nerves (e.g. hypoglossal-facial nerve neurorrhaphy or hemihypoglossal-facial nerve anastomosis or facial nerve decompression). These articles have been excluded, as have articles involving the use of acupuncture or electrical stimulation therapy in the recovery from facial paralysis or in nerve regeneration.
The purpose of our article was to review preclinical studies of substances that promote axonal growth of facial nerves, while excluding studies of surgical and conservative treatments involving the connection of damaged nerves with other nerves for the recovery of facial paralysis in patients. Among the 31 selected articles, the most common feature was the ability of various neurotrophic factors generated during facial nerve injury to affect nerve regeneration or survival. Although the factors that play a role to nerve regeneration are unclear, several neurotrophic factors (e.g. NGF, BDNF, GDNF, NT-3, VEGF, HGF and IGF) secreted by damaged nerves or specific growth factors secreted by transplanted stem cells have been found to be effective in regenerating facial nerves. In most cases, histological examinations showed that this microenvironmental resulted in increased axonal diameter as well as increased thickness of myelination. This analysis of available research on facial nerve injury and regeneration represents the first step toward future therapeutic strategies.
- Q) Which is the main surgical technique used in the 31 articles selected? How many articles present stem cell transplantation? How many used neurotrophic factors? What is the most widely used animal model for these studies? Authors can decide to address this issue here or in the conclusion.
- A) This review paper describes potential therapeutic strategies and substances for facial nerve regeneration based on preclinical studies. Most surgical methods can be summarized by three techniques.
In the first, one side of the facial nerve is exposed, followed by selection of the branch of the main trunk or facial nerve, performing anastomosis after full-thickness incision, and injecting a specific substance intraperitoneally or topically.
The second technique involves exposing one side of the facial nerve, followed by application of a specific substance after compression injury to the main trunk of the facial nerve.
In the third technique, one side of the facial nerve is exposed, followed by selection of the branch of the main trunk or facial nerve, creating a nerve defect. This defective nerve is connected to a nerve or vein conduit or to an autograft filled with a specific substance that has been attached to the proximal and distal stumps of the nerve.
The 31 articles performed experiments in rabbits, guinea pigs, rats and mice, with rats used most frequently (20 studies). These included seven studies involving Sprague-Dawley rats, six each involving Wistar and Lewis rats, and one involving Fisher rats. Twelve articles described the effects of stem cell transplantation, 11 the effects of neurotrophic factors, two the effects of stromal vascular fractions, and six to the effects of medications. All of the above has been summarized in the Introduction section of our manuscript:
- Introduction
…
Of the 31 papers, 12 were related to stem cell transplantation, two to stromal vascular fractions, 11 to neurotrophic factors, and six to medications. Three surgical techniques are commonly used. In the first, one side of the facial nerve is exposed and the branch or main stem of the facial nerve is selected. Anastomosis is performed following full-thickness incision and injection of a specific substance intraperitoneally or topically. In the second technique, one side of the facial nerve is exposed, and a specific substance is applied after compression injury to the main trunk of the facial nerve. In the third technique, one side of the facial nerve is exposed, and the branch or main trunk of the facial nerve is selected to create a nerve defect. The nerve, vein conduit or autograft is subsequently filled with a specific substance and connected to the proximal and distal stumps of the nerve.
Chapter 2:
- Q) The following sentence need at least one reference: “In addition, among the numerous treatments for promoting facial nerve repair, the most widely used in current clinical practice are local interventions, such as nerve anastomosis, nerve transplantation, and nerve conduits containing added neurotrophic factors or neural stem cells.” My perplexity concerns the facts that conduits enriched with neurotrophic factors and neural stem cells are used in clinic.
- A) At present, substances administered to patients are inadequate for peripheral nerve regeneration. This review describes current ongoing preclinical research on nerve regeneration. The sentence has therefore been modified.
In addition, among the numerous treatments for promoting facial nerve repair, the most widely used in current clinical practice preclinical studies are local interventions, such as nerve anastomosis, nerve transplantation, and nerve conduits containing added neurotrophic factors or neural stem cells.
Chapter 3:
- Q) Check if the numbering of paragraphs and sub-paragraphs is ok. I suppose that something is wrong. Each stem cell type and each factor is discussed singularly. A transversal discussion with author’s opinion about the different stem cells treatment and factors effect is missing. Author’s experience could be reported. Authors can add this discussion point in this chapter or in the conclusion.
- A) Chapter 3 in this review is important, describing the substances involved in nerve regeneration. For clarification, however, Chapter 3 has been reorganized; substances related to nerve regeneration have been classified as stem cells, stromal vascular fraction, neurotrophic factors, and medications.
This review concludes by summarizing the effects of various potential therapeutic substances on facial nerve regeneration, as shown in previous studies, as well as the substances they have in common for nerve regeneration. Neurotrophic factors secreted by damaged nerves are important, as they collectively provide a microenvironment that supports the survival or regeneration of axons. In addition, the degree of remyelination after nerve injury was shown to determine the speed of recovery. Several recent studies have investigated the active regenerative effects of stem cell-based therapy, finding that these stem cells continue to proliferate even after being transferred to damaged nerve tissues and differentiate into necessary cell types under appropriate microenvironmental conditions.
- Q) Check lines from 208 to 214. Something went wrong with the formatting of the text.
- A) We have removed the incorrect formatting of the text.
Chapter 5:
- Q) Lines 391-393: The authors have to specify that these are not resident stem cells, but transplanted stem cells.
A). As suggested, we have changed this sentence to:
Also important are specific growth factors secreted by transplanted stem cells, including NGF, BDNF, GDNF, NT-3, VEGF, HGF and IGF, which act to promote the recovery and regeneration of damaged peripheral facial nerves.
- Q) References are missing in the entire paragraph. For example, the sentence “A number of recent studies have investigated active regenerative effects of stem cell-base therapy, showing that these stem cells continue to proliferate even after being transferred to damaged nerve tissues and differentiate into necessary cell types under appropriate microenvironmental conditions.” need a reference. As well as for the sentence “histological examinations performed following transplantation of different stem cells (e.g., ADSCs, BMSCs) have revealed increases in the thickness of myelination as well as axonal dimeters.”
- A) As suggested, we have cited the appropriate references.
A number of recent studies have investigated active regenerative effects of stem cell-base therapy, showing that these stem cells continue to proliferate even after being transferred to damaged nerve tissues and differentiate into necessary cell types under appropriate microenvironmental conditions. [20, 99] Importantly, histological examinations performed following transplantation of different stem cells (e.g., ADSCs, BMSCs) have revealed increases in the thickness of myelination as well as axonal dimeters. [38, 53, 100]
Table:
- Q) The authors have to reorganize the table because, as it presented now, it is impossible to consult it. I suggest dividing the table in different tables, for example grouping articles that use the same technique of repair, cell treatment or therapeutic molecules. In the review, tables have to be cited.
- A) As suggested, the table would be more readable if articles were grouped by therapeutic substances. We have therefore reorganized the table into four tables and cited the appropriate references in the manuscript.
References:
References from 84 to 95 are not cited in the text but only in the table. I think that, if the authors selected these articles as important to be reported, they have to comment them in the manuscript.
- A) The table has been reorganized and the appropriate references cited.
Reviewer 2 Report
Facial nerve (VII cn) regeneration is a really important topic. The authors have done a complete review about this field of work. It could be useful for the paper and for readers to summarize in a tab the main neurological and surgical diseases in which it could be possible to have VII cn injury. Another point is to summarize surgical strategies in another tab before start with the list of pre clinical studies.
Author Response
* Response to Reviewer 2 comments
- Q) Facial nerve (VII cn) regeneration is a really important topic. The authors have done a complete review about this field of work. It could be useful for the paper and for readers to summarize in a tab the main neurological and surgical diseases in which it could be possible to have VII cn injury. Another point is to summarize surgical strategies in another tab before start with the list of pre clinical studies.
- A) The Introduction section of the manuscript now includes a list of the main neurological and surgical diseases that can cause facial nerve damage.
- Introduction
Peripheral facial palsy is caused by a variety of disease pathologies, including congenital conditions, Bell’s palsy, Ramsay Hunt syndrome, Guillain-Barre syndrome, Lyme disease, HIV infection, Kawasaki disease, central nervous system disorders, temporal bone fracture, otitis media, Melkersson-Rosenthal syndrome, neoplasms, and malignant tumors of the parotid gland and temporal bone.[1] The facial nerve follows a long and complicated course from the brainstem to the facial musculature. Most otological procedures, as well as parotid and facial surgeries, are centered around the facial nerve, making it vulnerable to iatrogenic injuries.[2] The most common iatrogenic facial nerve injury by sugery was temporomandibular joint replacement, and oral and maxillofacial surgical procedures accounted for 40% of injuries, resections of head and neck lesions 25%, otologic procedures 17%, cosmetic procedures 11%, and other procedures 7%.[3]
In addition, Reviewer 1 suggested that the Introduction section should summarize surgical strategies in facial nerve regeneration. These strategies have therefore been included, prior to the description of preclinical studies.
- Introduction
…
After application of exclusion criteria, a total of 31 qualifying studies were identified, and their methods and results were summarized (Figure 1). Of a total of 31 papers, 12 were related to stem cell transplantation, 2 were related to stromal vascular fraction, 11 were neurotrophic factor, and 6 were related to medication. There are three commonly used main surgical techniques as follows: Expose one side of the facial nerve and select the branch or main stem of the facial nerve.
- selecting the branch of the main trunk or facial nerve, performing anastomosis after full-thickness incision, and injecting a specific substance intraperitoneally or topically.
- Exposing one side of the facial nerve, a specific substance is applicated after compression injury to the main trunk of the facial nerve.
- Exposing one side of the facial nerve, selecting the branch of the main trunk or facial nerve, to create a nerve defect and connect nerve or vein conduit or autogaft filled with a specific substance to the proximal and distal stump of the nerve.
Round 2
Reviewer 1 Report
Authors answered adeguately to all points that I have rised, but Table 3 is not cited and all tables are missing in the last version of the manuscript.